# New Habits of Travellers Deriving from COVID-19 Pandemic: A Survey in Ports and Airports of the Adriatic Region

**Enrico Mancinelli** [1,*], **Umberto Rizza** [2], **Francesco Canestrari** [3], **Andrea Graziani** [3], **Simone Virgili** [1] **and Giorgio Passerini** [1]

1    Department of Industrial Engineering and Mathematical Sciences, Università Politecnica delle Marche, 60131 Ancona, Italy; s.virgili@staff.univpm.it (S.V.); g.passerini@univpm.it (G.P.)
2    Institute of Atmospheric Sciences and Climate, National Research Council, 73100 Lecce, Italy; u.rizza@isac.cnr.it
3    Department of Construction, Civil Engineering and Architecture, Università Politecnica delle Marche, 60131 Ancona, Italy; f.canestrari@staff.univpm.it (F.C.); a.graziani@staff.univpm.it (A.G.)
*    Correspondence: e.mancinelli@univpm.it

**Abstract:** The aim of this paper is to analyse the variations in the habits and the modes of transport of travellers departing from airports and ports during the COVID-19 outbreak. In the second year of the pandemic (i.e., from August to October 2021), travellers were invited to take part in an anonymous online survey at the terminal buildings of nine Italian and Croatian airports and ports located in the Adriatic region. Around 73% of respondents used public transport when travelling in the pre-COVID-19 period, whereas the corresponding share of respondents in the COVID-19 period was less than 50% and approximately 56% in the post-COVID-19 future. The main reason for not travelling by public transport was related to personal or sanitary reasons in the time of COVID-19. During the pandemic, around 39% of travellers preferred their own vehicle to public transport for moving to/from the airports and ports because of safety and sanitary reasons. With the pandemic, health was the main reason behind daily choices for up to 49% of the respondents. Moreover, a similar share of travellers considered health when choosing the transportation mode during and after the pandemic.

**Keywords:** public transport; mobility; travellers' preferences; tourists; business travellers; airport surface access; land access to ports; outbreak of SARS-CoV-2 virus

## 1. Introduction

The outbreak of the coronavirus disease 2019 (COVID-19) pandemic has posed a serious health risk to the population around the world, with a dramatic impact on daily life.

For the containment of the COVID-19 pandemic, several governments have adopted control measures such as quarantine, social distancing, travel restrictions, and contact precautions. According to De Vos [1], the potential implications of social distancing on daily travel patterns could cause a decrease in the demand for travel in general and specifically for public transport services.

Great attention has been paid to the effects of the coronavirus pandemic on travellers' attitudes and behaviour to help mobility managers to reshape the offer in the mobility sector and meet new mobility patterns. Efficient public transportation systems aim at discouraging the use of private cars, providing travellers with fast, environmentally friendly, and cost-effective solutions. However, sanitary and safety measures (e.g., physical distancing) imposed by the pandemic challenge the very concept of mass public transportation in urban areas [2].

In the EU, transport demand is likely to decrease in the number of trips, average trip distance, and the use of public transport in the short term following the COVID-19 pandemic, whereas the evolution of the mobility patterns depends not only on the demand but also on the supply side in the medium term [3]. The main findings reported

by Lozzi et al. [4] regarding public transport services in European urban areas during and immediately after the lockdown were (i) the use of public transport and shared mobility services decreased dramatically, whereas the use of private vehicles such as cars and bicycles increased, as well as walking; (ii) the risk of contagion likely discouraged the use of shared mobility services.

The decision-making process is subject to both rationality and emotions. According to Lamb et al. [5], during COVID-19 and in the post-COVID-19 future, the willingness to fly among both business and leisure travellers could be predicted with the same set of variables, namely perceived COVID-19 threat, agreeableness, affect, and fear. Investigating the relationship between COVID-19 perception, travel risk perception, and behaviour among travellers, Neuburger and Egger [6] observed higher travel risk perception for females, older people, and those with lower travel frequency. Tarasi et al. [7] have reported differences between the factors affecting travel mode choice for men and women, with women considering personal safety, road safety, and ecological footprint more important than men. On the contrary, a mindfulness attitude to moment-by-moment experiences may be the key for coping with stressful situations in everyday activities [8] or extraordinary situations such as travelling for tourism [9]. Arnsten et al. [10] have pointed out the underlying mechanism during chronic stress such as during a pandemic outbreak, with the amygdala becoming more active, impairing the pre-frontal cortex and its decision-making and cognitive control processes. Readers can refer to the literature overviews reported in the works by Šinko et al. [11] and Shakibaei et al. [12] about changes in travel behaviour related to the COVID-19 pandemic.

With the COVID-19 outbreak, a preference for private cars compared to public transport emerged in studies performed in Germany [13,14], the United States [15], and India [16]. Comparing modal choices between pre-COVID-19 (2019) and the COVID-19 situation (2021), Šinko et al. [11] observed that a large percentage of areas in Vienna (Austria) shifted from public transport to automobiles (around 44%) or bicycles (around 12%). In Italy, analysis of the impact of COVID-19 measures on Rome's mobility system showed a more rapid return to trips by car compared to public transport [17]. However, in Italy, the modal split of private vehicles was roughly the same as the pre-pandemic levels in the five-month period following the 2020 lockdown [18].

Previous works have investigated changes in travellers' habits due to the COVID-19 outbreak, with surveys restricted to urban areas [7,11,12,19,20], nationwide [14,16,18,21–23], and worldwide [24]. However, the respondents were generally collected for purposes such as representativeness of the population for the area under study. For example, several studies located in urban areas [7,11,12,19,20,25] performed surveys to analyse the travel behaviour of daily commuters during the COVID-19 pandemic. Therefore, there is a research gap in investigating the variations in the modes of transport of travellers departing from airports and ports during the COVID-19 pandemic.

The aim of this paper is to analyse the variations in the modes of transport of travellers departing from airports and ports in the Adriatic region deriving from the COVID-19 outbreak.

In the second year of the pandemic (i.e., from August to October 2021), an anonymous online survey was distributed at the terminal buildings of nine Italian and Croatian airports and ports located in the Adriatic region. Questions referred to the post-pandemic near-future, pre-pandemic, and pandemic periods to understand any changes in the habits of leisure and business travellers related to transportation. The survey was held within the Green and Intermodal Solutions for Adriatic Ports and Airports "ADRIGREEN" project [26], aimed at studying sustainable mobility solutions for ports and airports. While approaching the research on mobility within the ADRIGREEN project, the outbreak of the pandemic has produced disruptions in the habits of travellers. This prompted us to investigate the changes in travellers' habits regarding modes of transport.

The remainder of this paper is organised as follows: Section 2 describes data and calculations utilised for this study, including the structure of the survey, study area, and

sample characteristics; in Sections 3 and 4, results and discussions are presented; finally, in Section 5, the conclusions of the study are drawn.

## 2. Data and Calculations

### 2.1. Structure of the Survey

The target population of the survey consisted of adult travellers departing from ports and airports in the second year of the pandemic (i.e., from August to October 2021). Travellers were invited to take part in an anonymous online survey by using a quick response code or digiting a link, both printed on flyers. The flyers had been distributed by the airport and port authorities at the holding areas and boarding gates of passenger terminals.

The survey was designed by means of Google Forms software in English and Italian languages. The participants were informed about the scientific purposes of the survey.

The survey was structured in five sections, with the following topics: (i) travellers' perceptions about the risk of COVID-19 infection; (ii) travellers' habits regarding the usage of their own vehicle, public transport, or other means of transportation in different contexts (e.g., when travelling or during the stay); (iii) travellers' opinions about various means of transportation; (iv) travellers' awareness about health and environmental issues; (v) sociodemographic information and general information about travel frequency and purpose.

From Section 2 to Section 4, each question referred to the post-pandemic near-future, pre-pandemic, and pandemic periods. Therefore, the respondents had to recall or foresee their habits in the pre-pandemic period or post-pandemic future. This may represent a limitation to the representativeness of the survey. However, there are a few studies investigating changes in modal split based on travellers' surveys performed before and during the COVID-19 outbreak [19]. The relatively small sample size represents another limitation of the survey. The answer options were checkboxes or single or multiple choices.

### 2.2. Examined Study Area

The travellers' survey was conducted at small- to medium-sized ports and airports serving the Adriatic basin, namely four Croatian airports and ports (Dubrovnik and Pula), four Italian airports (Bari, Brindisi, Pescara, and Rimini), and one Italian port (Ancona) (Figure 1).

From August to October 2021, the total passenger traffic at the airports of Bari, Brindisi, Pescara, and Rimini accounted for around 7% of the whole passenger traffic in Italian airports (Table 1). The monthly total traffic of passengers at Dubrovnik and Pula airports accounted for between 24.5 and 30.3% of the whole air passenger traffic in Croatia (Table 1). Most of the small- to medium-sized Croatian and Italian airports serving the Adriatic region rely primarily on passenger traffic and experience peaks in the summer season [27].

The total passenger traffic at the Dubrovnik and Pula ports was up to around 12% of the whole passenger traffic in Croatian ports (Table 2). No data were available about total passenger traffic for the port of Ancona.

To evaluate the effect of the pandemic on the volume of passenger traffic at the Croatian and Italian airports and ports, passenger traffic from August to October 2021 (during the pandemic) was compared with the values from August to October 2019 (prior to the pandemic) as follows:

$$Percentage\ change_i = \frac{(Passengers_i\ 2021 - Passengers_i\ 2019)}{Passengers_i\ 2019} \times 100 \qquad (1)$$

*Percentage change$_i$*—percentage change in passenger traffic in *i*-month (with *i* = 8, 9, and 10); *Passengers$_i$*—volume of passenger traffic in *i*-month.

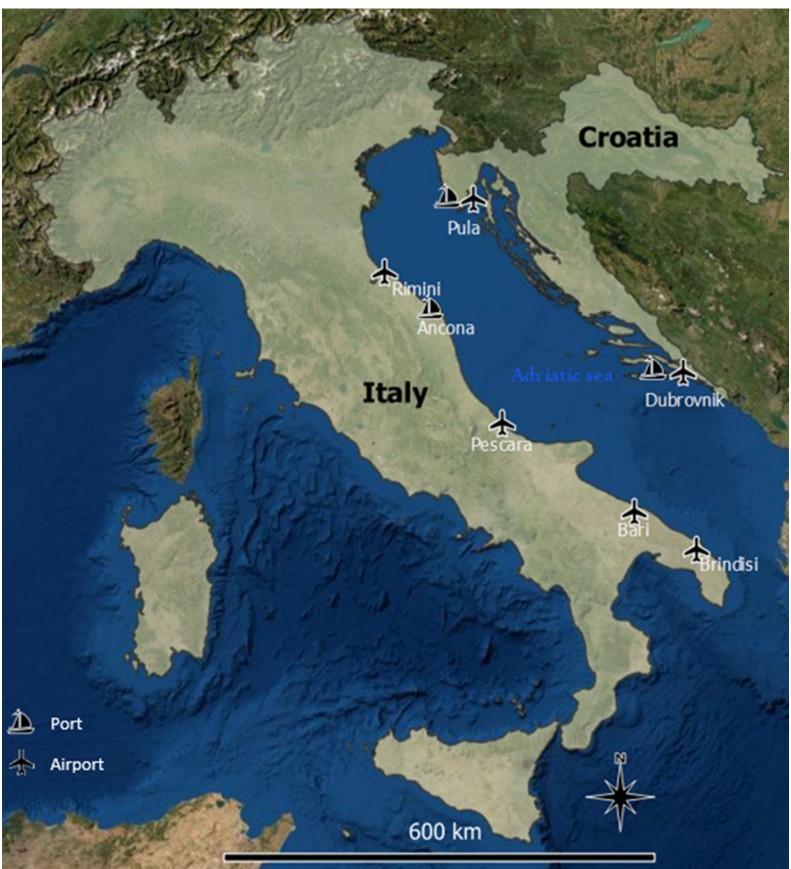

**Figure 1.** Location of the Croatian and Italian airports and ports in the Adriatic region.

**Table 1.** Total (national and international) passenger traffic in COVID-19 period (i.e., from August to October 2021) and percentage change in total passenger traffic between pre-COVID-19 (2019) and COVID-19 periods (2021) in the airports of Bari, Brindisi, Pescara, Rimini, Dubrovnik, Pula, Croatia, and Italy. Authors' own elaboration based on data from Assaeroporti [28] and Croatian Bureau of Statistics [29].

| Airport Passenger Traffic | August 21 [–] | September 21 [–] | October 21 [–] | Percentage Change in August [%] | Percentage Change in September [%] | Percentage Change in October [%] |
|---|---|---|---|---|---|---|
| Bari | 531,325 | 480,518 | 463,267 | −10.7 | −13.2 | −8.0 |
| Brindisi | 321,063 | 280,132 | 246,326 | 8.1 | 0.9 | 1.0 |
| Pescara | 68,748 | 57,001 | 55,970 | −4.0 | −16.8 | −6.5 |
| Rimini | 14,900 | 17,215 | 12,893 | −77.7 | −71.2 | −62.1 |
| Italy | 13,009,114 | 11,377,934 | 10,627,995 | −36.2 | −39.7 | −37.1 |
| Dubrovnik | 288,466 | 206,825 | 114,346 | −44.8 | −48.7 | −61.6 |
| Pula | 90,187 | 58,916 | 23,652 | −48.3 | −52.1 | −45.4 |
| Croatia | 1,250,165 | 912,057 | 563,750 | −35.9 | −39.0 | −44.5 |

In pandemic period, the whole air passenger traffic was far lower than the pre-pandemic levels both in Croatia and Italy (Table 1).

**Table 2.** Passenger traffic (i.e., departures and arrivals) in COVID-19 period (i.e., from August to October 2021) and percentage change in passenger traffic between pre-COVID-19 (2019) and COVID-19 periods (2021) in the ports of Dubrovnik, Pula, and Croatia. Authors' own elaboration based on data from Croatian Bureau of Statistics [29].

| Port Passenger Traffic | August 21 [–] | September 21 [–] | October 21 [–] | Percentage Change in August [%] | Percentage Change in September [%] | Percentage Change in October [%] |
|---|---|---|---|---|---|---|
| Dubrovnik | 375,992 | 244,421 | 111,105 | −43.4 | −44.3 | −65.7 |
| Pula | 208,174 | 116,843 | 98,691 | 6.7 | −3.2 | 6.9 |
| Croatia | 6,753,711 | 3,481,863 | 1,798,632 | −15.8 | −14.0 | −25.1 |

In the COVID-19 period, passenger traffic was between 14 and 66% lower than the pre-pandemic levels in Croatian ports (Table 2). Differences were noted for the volumes of total monthly passenger traffic, with the airports of Rimini, Dubrovnik, and Pula and the port of Dubrovnik registering values between 43 and 78% lower than the pre-COVID-19 levels, whereas Brindisi airport and Pula port showed values up to 8% higher than the pre-pandemic levels (Tables 1 and 2). The airports of Bari and Pescara showed values comparable to pre-pandemic levels (Table 1).

Table 3 reports the locations of the airports and ports and the public means of transport serving the infrastructures. The ports of Ancona and Pula are at walking distance from the train stations serving the cities. The only airport with a train station at the terminal is Bari, whereas Rimini airport is within walking distance of the train station.

**Table 3.** Locations of airports and ports and types of public transport serving the infrastructure.

| | Region | Country | Distance from Nearby City (km) | Public Means of Transport |
|---|---|---|---|---|
| Ancona port | Marche | Italy | <1 | Bus, train station (distance < 2 km) |
| Bari airport | Apulia | Italy | 9 | Bus, taxi, car rental, train station (at the terminal) |
| Brindisi airport | Apulia | Italy | 5 | Bus, taxi, car rental |
| Dubrovnik airport | Adriatic Croatia | Croatia | 22 | Bus, taxi, car rental |
| Dubrovnik port | Adriatic Croatia | Croatia | 2 | Shuttle bus, taxi |
| Pescara airport | Adriatic Croatia | Italy | 4 | Bus, taxi, car rental |
| Pula airport | Adriatic Croatia | Croatia | 6 | Bus, taxi, car rental |
| Pula port | Adriatic Croatia | Croatia | <1 | Bus, taxi, train station (distance around 1 km) |
| Rimini airport | Emilia-Romagna | Italy | 8 | Bus, taxi, car rental, train station (distance around 1 km) |

### 2.3. Sample Characteristics

The average time required to complete the questionnaire was around 7.9 min. Of the 117 questionaries that were filled in online, 85 questionaries were considered for analysis. Total response time was considered for screening the survey answers. Questionnaires completed in under 3.4 min (i.e., first quartile) or over 30 min were discarded. Inaccurate answers may result from a low amount of time spent on average on each survey item [30]. A long total response time was considered a proxy measure of respondent burden [31], with potential implications on non-response or careless response rates [32].

Approximately half of the respondents were men (Table 4), travelling for holidays or leisure with a partner or a friend 1-2 times per year (Figure 2). The age group that was most

represented was people aged 21 to 30 (around 38% of the respondents), followed by people aged 31 to 40 and 41 to 50, with around 22% of the respondents in each group (Table 4). Most of the respondents (around 65%) had a university degree (Figure 2).

**Table 4.** Socio-demographic characteristics of the respondents.

| | | Number of Respondents | [%] |
|---|---|---|---|
| Gender | Female | 36 | 42.4 |
| | Male | 42 | 49.4 |
| | Do not want to specify/no answer | 7 | 8.2 |
| Age | <20 | 1 | 1.2 |
| | 21–30 | 32 | 37.6 |
| | 31–40 | 19 | 22.4 |
| | 41–50 | 19 | 22.4 |
| | 51–60 | 7 | 8.2 |
| | 61–70 | 2 | 2.4 |
| | Do not want to specify/no answer | 5 | 5.9 |
| Citizenship | Croatian | 45 | 52.9 |
| | Italian | 12 | 14.1 |
| | European Union | 17 | 20 |
| | Outside European Union | 3 | 3.5 |
| | Do not want to specify/no answer | 8 | 9.4 |
| Education | No high school | 3 | 3.5 |
| | High school or equivalent | 17 | 20 |
| | University degree | 55 | 64.7 |
| | Do not want to specify/no answer | 10 | 11.8 |

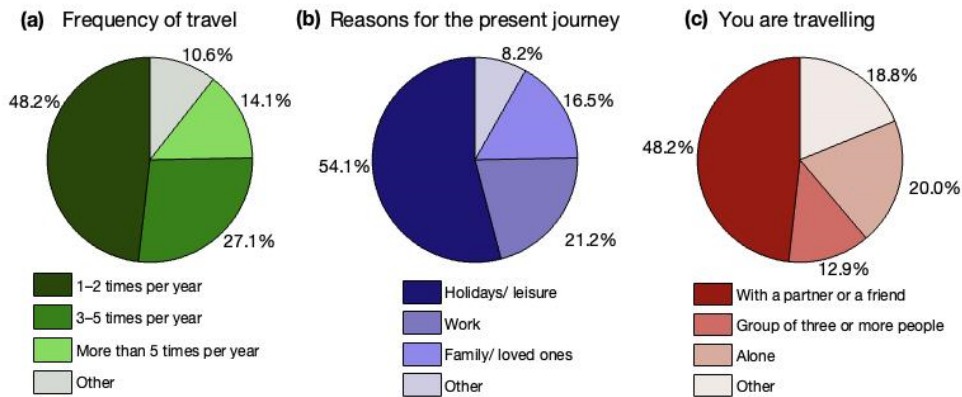

**Figure 2.** Respondents' frequency of travel per year (**a**), reasons (**b**), and (**c**) company for the present journey.

## 3. Results

### 3.1. Perception of the Risk of COVID-19 Infection When Travelling

To feel protected from the risk of COVID-19 infection when travelling, more than 50% of the respondents considered the two options of being vaccinated and physical distancing and face masks to be valuable (Figure 3a). However, around 32% of the travellers stated that they would feel protected by avoiding crowded areas and knowing that nearby people had been vaccinated or tested (Figure 3a). These conditions are difficult to fully implement and do not promote travellers' confidence. Therefore, it is likely that a relatively high percentage of respondents were travelling with low or no confidence. For this segment of travellers, measures such as "COVID-19-free tourist travel corridors" [33] or COVID-19-free islands [34] could be a way to restore travellers' confidence. On the other hand, around 49% of the respondents had no fears about COVID-19 when travelling by public means of transport (Figure 3b), whereas up to around 48% of the respondents sometimes did not feel protected.

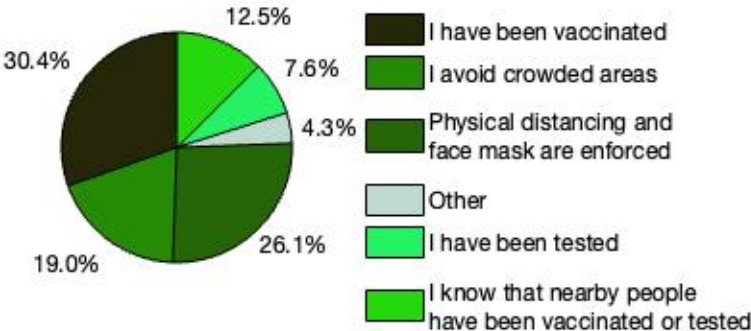

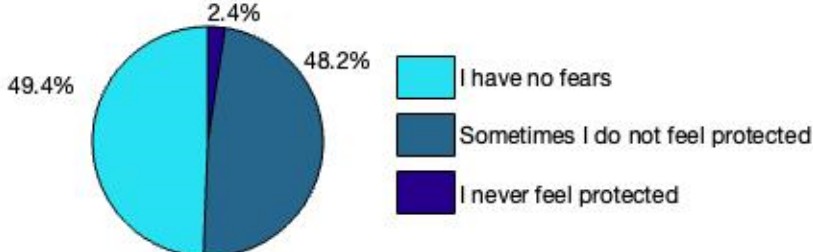

**Figure 3.** Preferred measures for feeling protected from COVID-19 infection when travelling in general (**a**), and perception of the risk of COVID-19 infection when travelling by public means of transport (**b**).

### 3.2. Travellers' Use of Public Transport, and Preferences for the Location and Transport Mode during the Stay

The majority (around 73%) of respondents used public transport when travelling in the pre-COVID-19 period, whereas the corresponding share of respondents in the COVID-19 period was 46% (Figure 4a). In the post-COVID-19 future, around 56% of travellers are likely to use public transport. The main reason for not travelling by public transport in the COVID-19 period was related to personal or sanitary reasons (Figure 4a). Around 48% of the respondents were reported to travel for different times per journey by public transport in the COVID-19 period, whereas around 37% of the respondents considered travelling a few times by public transport in the post-COVID-19 future (Figure 4a).

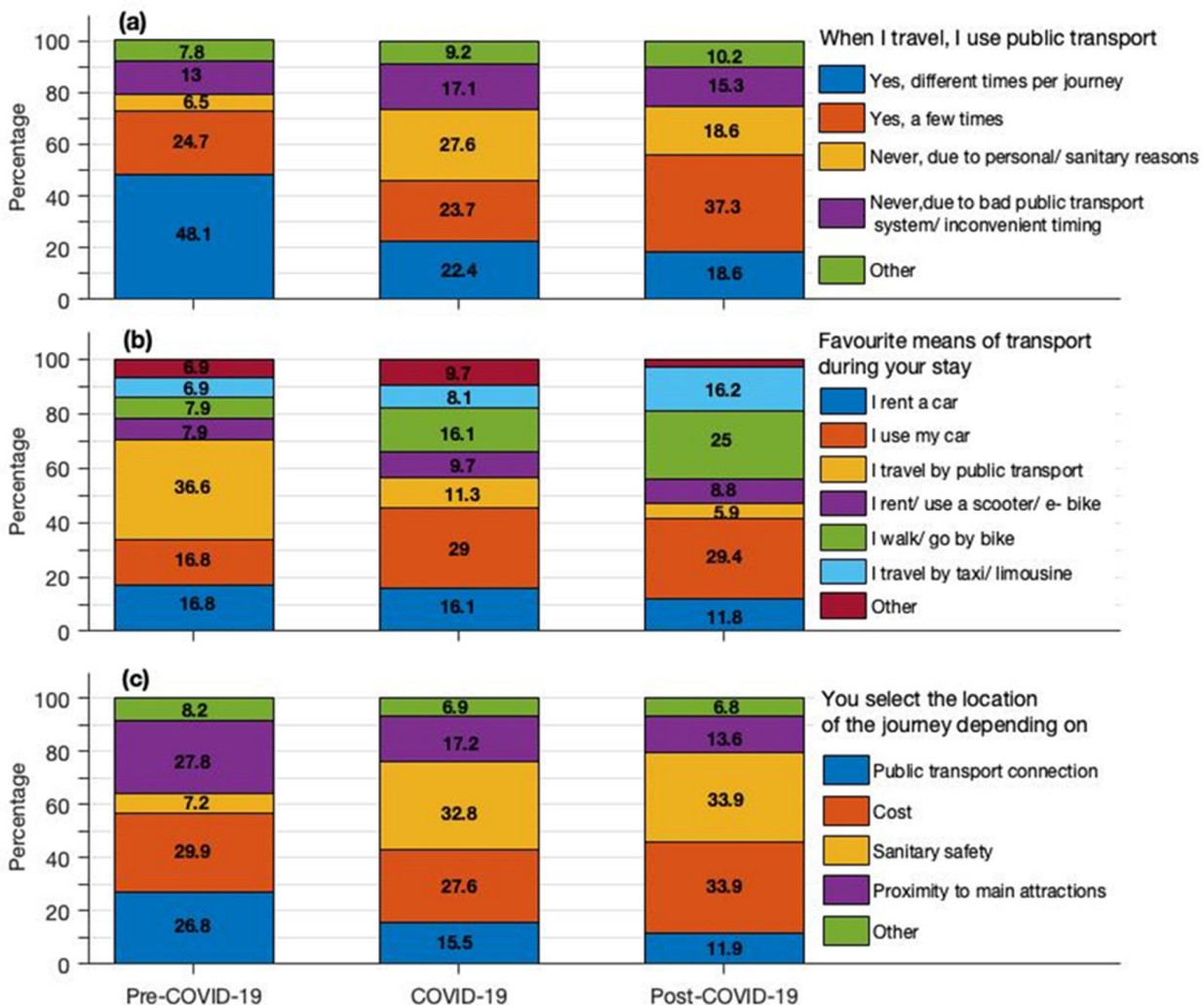

**Figure 4.** Comparisons of the travellers' preferences for (**a**) public transport when travelling, (**b**) the means of transport during the stay, and (**c**) the location of the journey during pre-, post-, and COVID-19 periods.

In the pre-COVID-19 period, a preference for the car or public transport was expressed by around 70% of travellers for moving around during their stay (Figure 4b). During the pandemic, the car was the preferred mode of transport for around 45% of the travellers, followed by walking or cycling (16% of the travellers).

In the post-COVID-19 future, most of the travellers are likely to move around by car (41%) or walking and cycling (25%) (Figure 4b). Comparisons between pre- and post-COVID-19 periods show a shift from public transport to car, walking and cycling, or taxi or limousine use.

Regardless of the pandemic, cost is one of the reasons for the choice of the location of the journey (Figure 4c). Travellers also selected the location of the journey depending on public transport connections and proximity to main attractions in the pre-COVID-19 period, whereas sanitary safety factors influenced travellers' destinations in the COVID-19 and post-COVID-19 periods (Figure 4c). Following the COVID-19 lockdown, sanitary conditions were among the key factors for selecting the location of the journey (DNA 2020 cited by Marques Santos et al. [35]).

### 3.3. Travellers' Habits and Reasons for Using Public or Private Means of Transport

Regardless of the pandemic, between 20 and 25% of the respondents used public transport to travel to/from airports and ports (Figure 5a). In the pre-COVID-19 period, the highest share of preferences was for 'sometimes' travelling by public transport to/from the airports and ports, for around 40% of the respondents. With the pandemic, the highest share of preferences was for 'never' travelling by public transport to/from the airports and ports, with around 39% (Figure 5a). Only 25% of respondents answered 'sometimes' to the question 'I will travel by public transport' in the post-COVID-19 period, compared to pre-pandemic (around 40% of respondents) (Figure 5a).

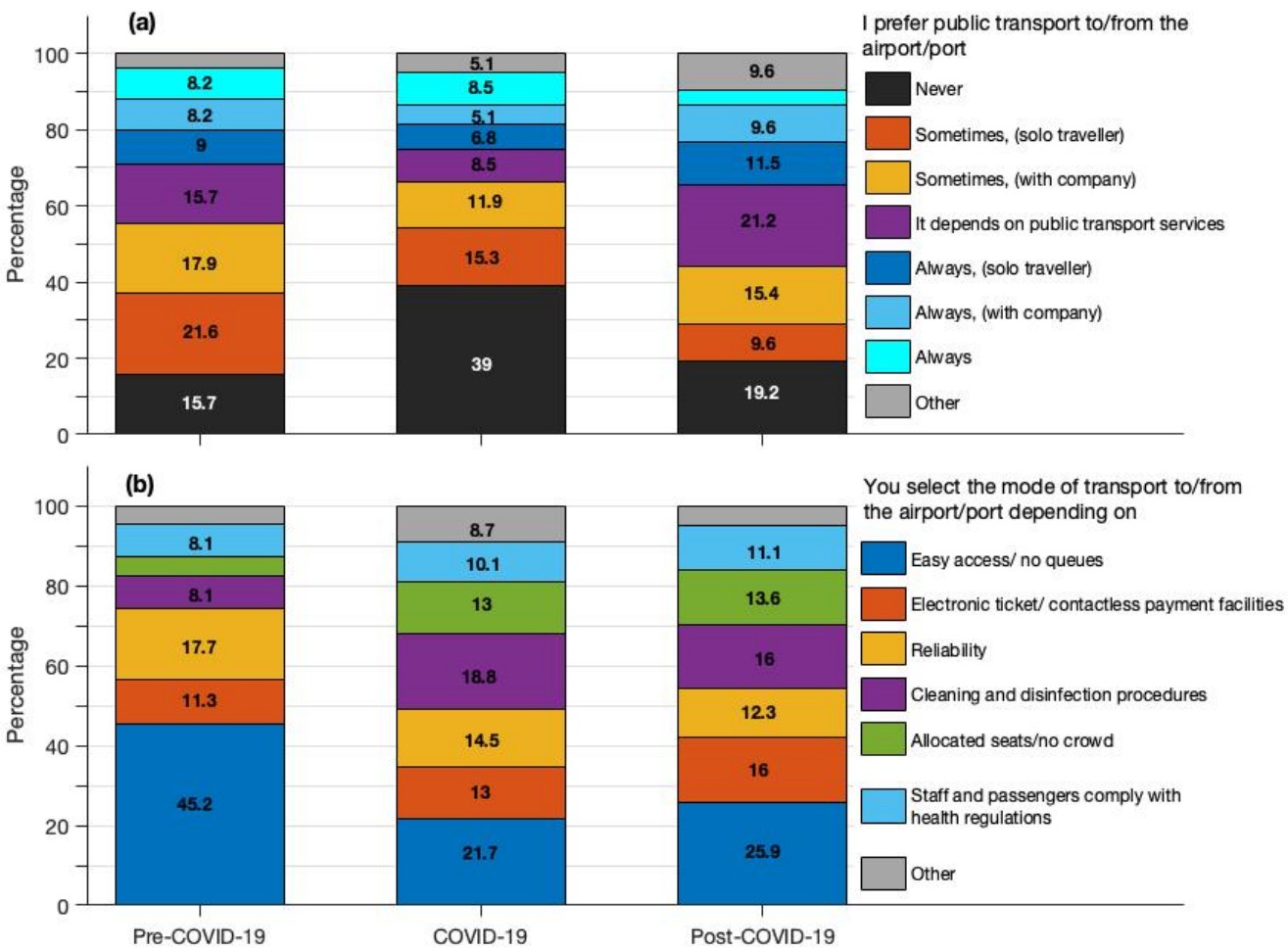

**Figure 5.** Comparisons of the travellers' habits for public means of transport (**a**), and reasons for travelling (**b**) by public means of transport to/from the airport/port during pre-, post-, and COVID-19 periods.

More travellers considered the option of travelling by public transport in the post-COVID-19 period (21% of respondents) compared to the pre-pandemic (around 16% of respondents) and pandemic periods (around 9% of respondents) depending on the transport services (Figure 5a).

A slightly higher percentage of respondents claimed that they were never going to travel by public transport to/from airports and ports in the post-COVID-19 (19% of respondents) compared to the pre-pandemic period (around 16% of respondents) (Figure 5a).

In the pre-COVID-19 period, easy access/no queues were the main reasons for selecting the mode of transport to/from airports and ports, with around 45% of preferences (Figure 5b). With the pandemic, there was no longer a clear expression of preferences for selecting the mode of transport, with easy access/no queues (around 22% of preferences)

and cleaning and disinfection procedures (around 19% of preferences) being the most preferred options (Figure 5b). The same trend is evident in the post-pandemic future.

The reasons for not travelling by public transport to/from the airports and ports (Figure 6a) were that the respondents considered feeling at risk due to poor safety/health conditions during (around 45% of preferences) and after (25% of preferences) the pandemic. The respondents did not express such a clear motive in the pre-COVID-19 period, with the highest shares (up to around 22%) of preferences for (i) frequency of connections or overcrowding and (ii) no public transport or poor time slots.

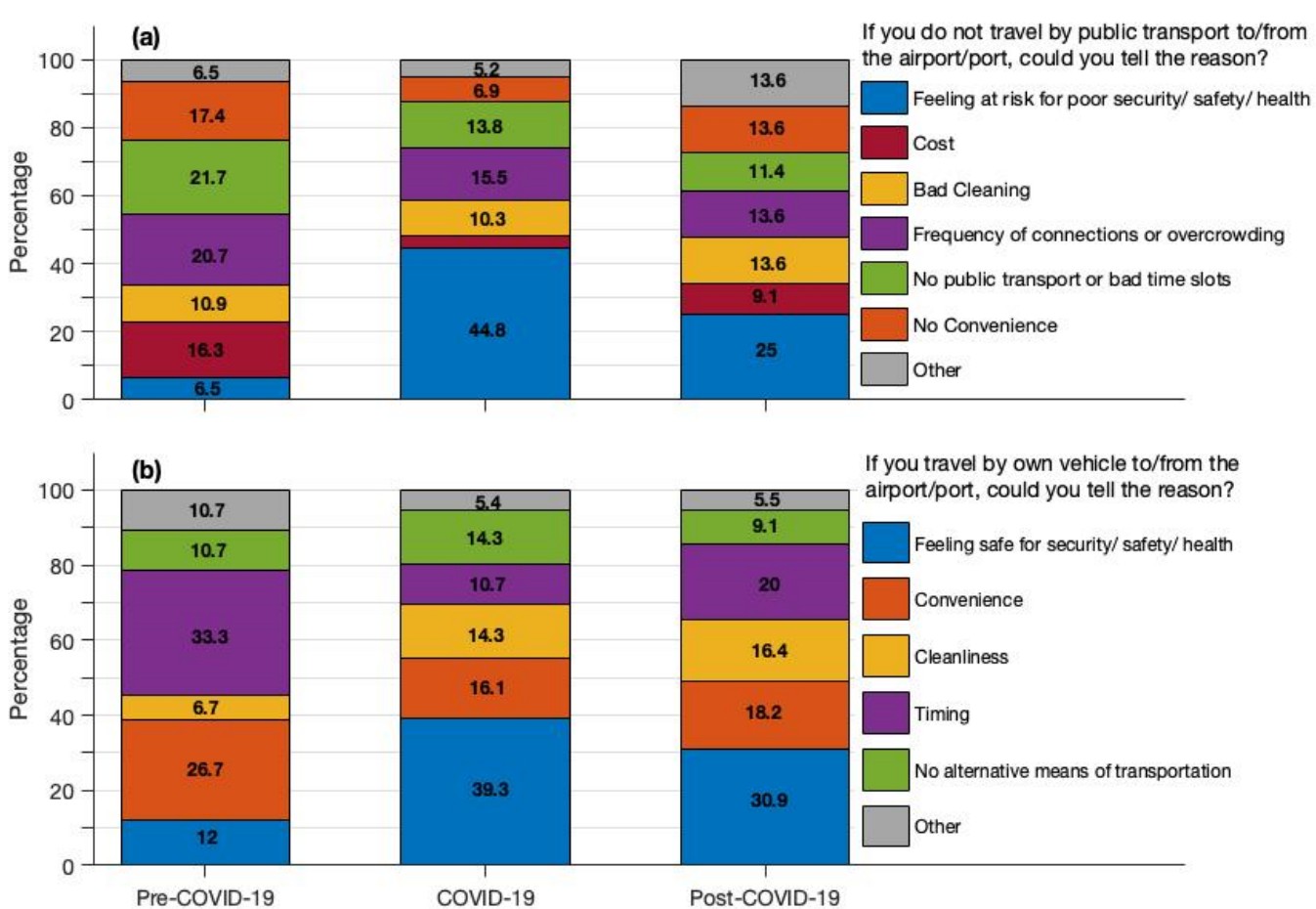

**Figure 6.** Comparisons of the travellers' reasons for not travelling (**a**) by public means of transport, and for travelling by (**b**) their own vehicle to/from the airport/port during pre-, post-, and COVID-19 periods.

In the pre-COVID-19 period, timing and convenience were the main reasons for travelling by one's own vehicle to/from the airports and ports (Figure 6b), with up to 60% of preferences. With the pandemic and in the post-pandemic future, up to 39% of respondents considered feeling safe regarding safety/health conditions when travelling by their own vehicle (Figure 6b).

### 3.4. Travellers' Opinions about Alternative Means of Transport, Health, and Environmental Awareness

Up to 55% of the respondents did not consider alternative means of transport (e-bike, etc.) useful to move to/from the airports and ports both before and during the pandemic, whereas the share for not considering alternative means of transport useful decreased to 27% of the respondents in the post-COVID-19 future (Figure 7a). Moreover, more travellers are likely to consider alternative means of transport depending on the

available options in the post-COVID-19 future (around 24% of respondents) compared to the pre-COVID-19 period (around 9% of respondents).

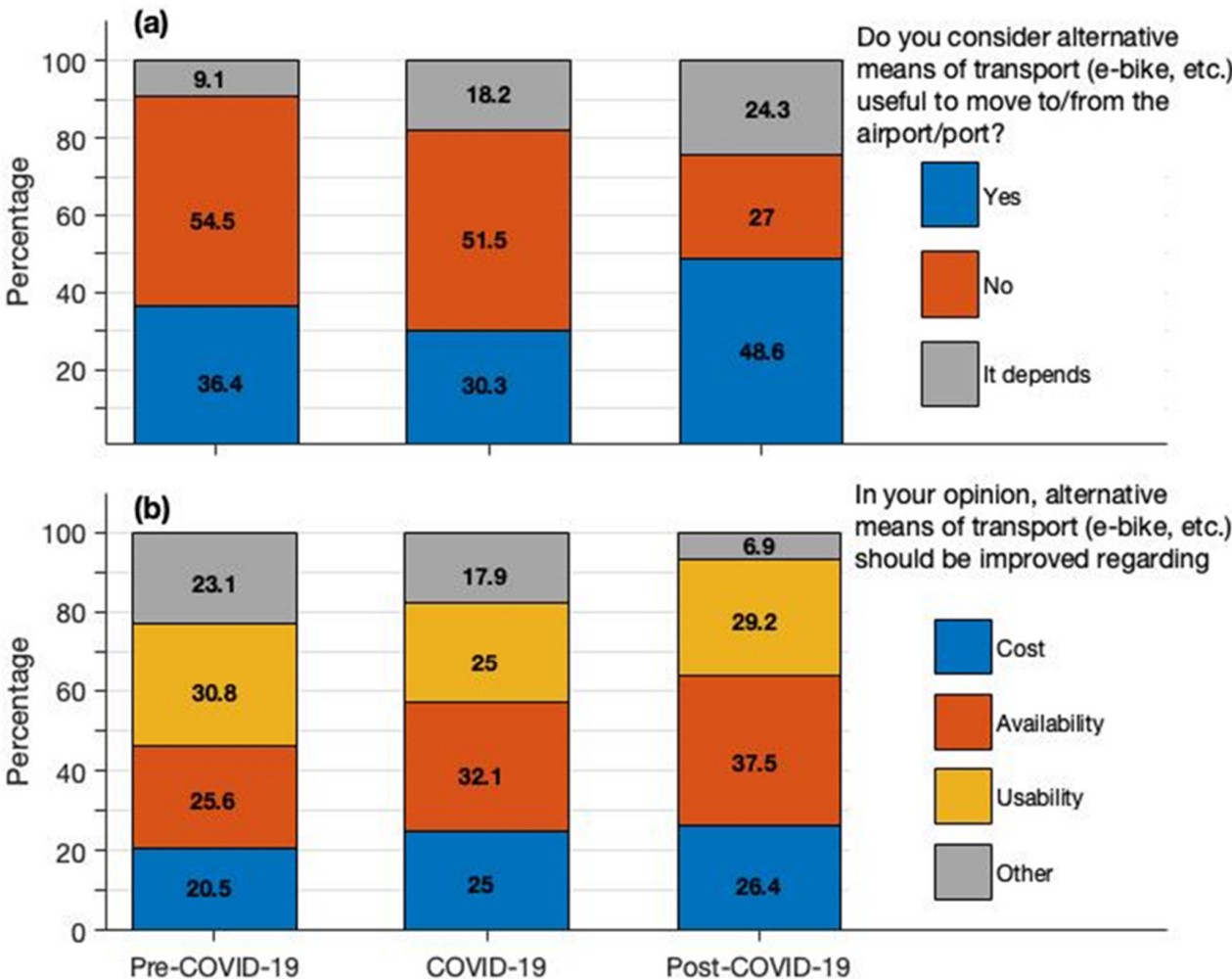

**Figure 7.** Comparisons of travellers' opinions about alternative means of transport to/from the airports and ports (**a**) and aspects to be improved (**b**) during pre-, post-, and COVID-19 periods.

Travellers were interviewed about the aspects of alternative means of transport that could be improved (Figure 7b). Both before and during the pandemic, preferences were roughly the same for cost, usability, and availability, with values between 20 and 30%. In the post-COVID-19 future, around 38% of travellers believed that the availability of alternative means of transport should be improved.

Before the pandemic, the respondents did not express a clear preference either for the issues influential to their daily choices (Figure 8a) or for the issues related to transportation (Figure 8b).

Health was the most influential issue on daily choices during the pandemic (49% of the respondents), and in the post-pandemic future (around 38% of the respondents) (Figure 8a).

The same issue was considered related to transportation during the pandemic (around 51% of the respondents) and in the post-pandemic future (around 29% of the respondents) (Figure 8b).

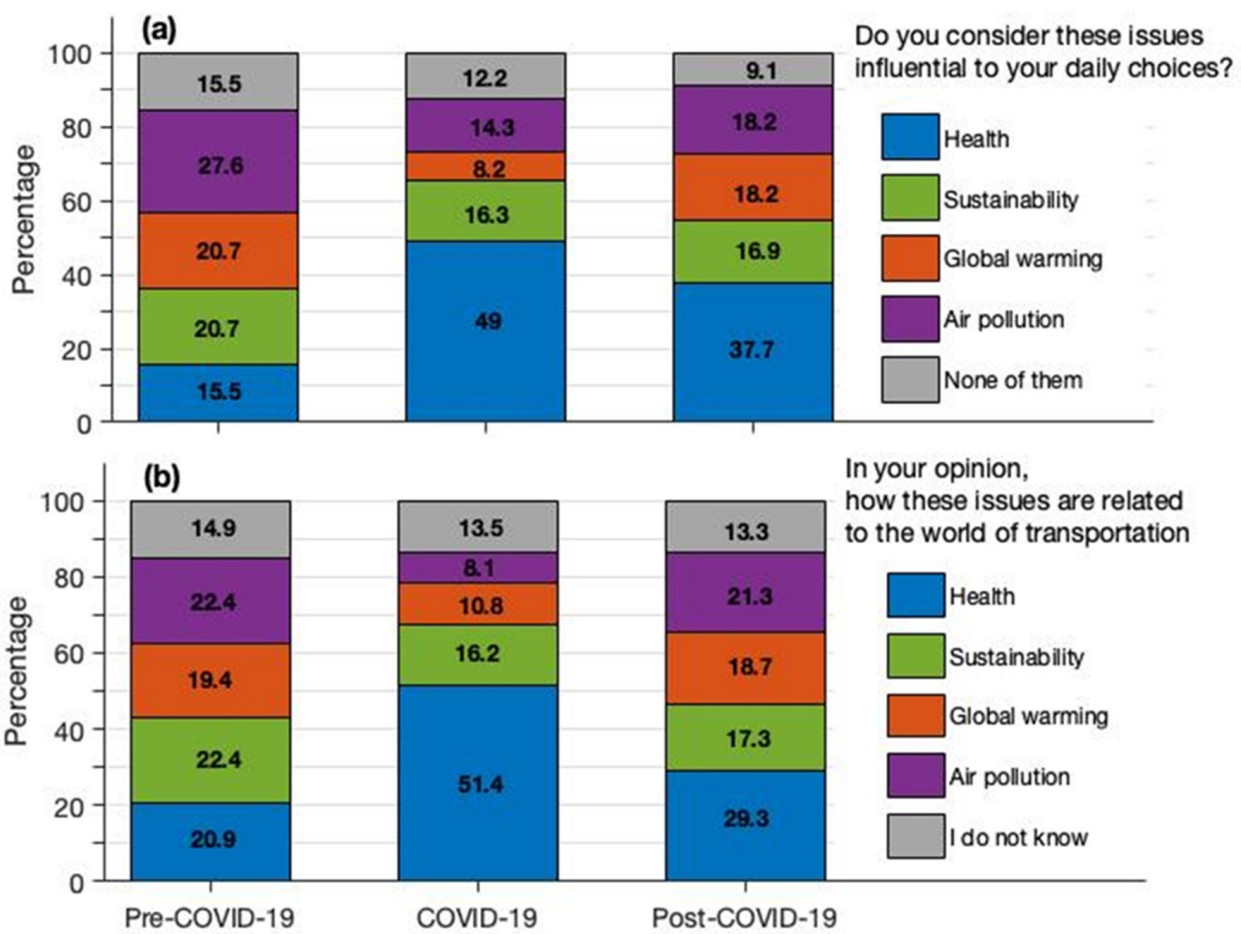

**Figure 8.** Comparisons of travellers' awareness of health and environmental issues related to (**a**) daily choices, and (**b**) transportation during pre-, post-, and COVID-19 periods.

## 4. Discussions

In terms of sample size and characteristics, it was more effective to advertise the survey via the distribution of flyers in specific areas of passenger terminals compared to the option of displaying posters in various areas of ports and airports. However, port and airport authorities had to allocate higher resources for distributing flyers compared to displaying posters, also for reducing the risk of COVID-19 contagion. Moreover, this survey method proved to be very useful given the ease of passengers to access the online survey via smartphones and other mobile devices. The survey was set up in such a way as to be able to separate the responses of each infrastructure. We recommend this methodology, which allows the extraction of additional statistical information. On the other hand, this required much more time during the design of the survey due to a certain organizational complexity, but it greatly simplified the post-processing of the data.

During the pandemic, around 40% of travellers preferred their own vehicle to public transport for moving to/from the airports and ports because of safety and sanitary reasons (Figure 6). Our findings are in line with previous studies [19,20] reporting that willingness to use public transport for commuting in urban areas is affected by factors such as safety perception and sanitary reasons in a pandemic situation.

With the pandemic, a shift from public to private modes of transport is likely to have higher external costs (i.e., the hidden costs for society, such as road maintenance, air pollution, etc.). For example, the average external costs of passenger transport by car were around 5.3 and 7.4 €-cents/(passenger × km) higher than by bus in Italy and Croatia, respectively [36]. A decrease in the volume of passenger traffic will likely act as a

buffer for the higher external costs related to a shift in passengers from public transport to private vehicles at the airports of Rimini, Dubrovnik, and Pula, and the port of Dubrovnik. However, Sifakis et al. [37] observed no correlation between monthly levels of airborne pollutants and the total number of tourists in a medium-sized seaside resort in Crete (Greece) during pre-, post-, and COVID-19 periods. Attention should be paid to the locations (e.g., Pula, Bari, Brindisi, and Pescara) served by airports and ports with volumes of passenger traffic higher than or comparable to the pre-COVID-19 levels (Tables 1 and 2).

To avoid/limit the higher external costs deriving from a shift from public to private transport, stakeholders may consider actions such as (i) promoting the use of alternative modes of transport, (ii) launching campaigns for restoring travellers' confidence in public transport, (iii) monitoring mobility behaviour and disseminating road transport externalities (e.g., ambient air quality and related health risks). Considering the European vision of sustainable mobility, the European Commission is planning to foster the internalisation of external costs of transport according to principles such as 'polluter pays' and 'user pays' [38].

For example, to overcome the impact of COVID-19 on urban mobility and to reduce the use of private and polluting vehicles, Lozzi et al. [4] suggested to encourage soft mobility, i.e., any human-powered (non-motorised) or small e-mobility modes (e.g., e-bike and e-scooter). During peak hours, public transport managers could consider diverting the demand for public transport services to other collective, shared, and sustainable mobility services, such as bike sharing [39]. However, between 27 and 55% of the respondents did not consider alternative means of transport useful to move to/from the airports and ports (Figure 7a). In this framework, soft mobility and micro-mobility may be a solution to avoid increasing the use of cars. However, such solutions are suitable for ports that are generally part of the urban conurbation (Table 3). On the contrary, it is unlikely that passengers at airports will choose these mobility solutions to commute to the nearby city or multimodal node. The offer of e-vehicles, car sharing, car-pooling, and charging stations would be an interesting option for developing low-carbon accessibility in airports. In the United Kingdom, different airports consider initiatives such as providing pedestrian routes and cycle paths between airports and nearby residential areas for discouraging the use of private cars among airport employees [40].

Moreover, public transportation services require the joint effort of port/airport management and local stakeholders on the supply side, as well as a more positive attitude towards public transport on the demand side. According to a survey reported in the 'European Aviation Environmental Report 2019' [41], almost all the airports (namely 98%) were served by public transportation systems. However, less than half of the airports' employees and travellers were reported to use public transport to reach the airport infrastructure [41].

With the pandemic, health was the main reason behind daily choices for around 30 to 50% of the respondents (Figure 7a). The same issue was considered related to transportation during the pandemic and after the pandemic by roughly the same share of respondents (Figure 7b). This is likely due to the association/connection between stay-at-home orders and the initial shock of the risk of contagion. The same concern for health regarding daily choices and mobility could be the starting point for boosting the transition to smart and sustainable modes of transport.

## 5. Conclusions

The aim of this paper is to analyse the variations in the habits and the modes of transport of travellers departing from airports and ports with the COVID-19 outbreak. Despite of the number of articles dealing with changes in travellers' behaviour with the COVID-19 pandemic, we observed a research gap regarding current and future trends of surface access to airports and ports estimated through surveys performed during the crisis.

In the second year of the pandemic (i.e., from August to October 2021), an anonymous online survey was performed at the terminal buildings of small- to medium-sized airports and ports serving the Adriatic region, namely four Croatian airports and ports (i.e., Dubrovnik

and Pula), four Italian airports (i.e., Bari, Brindisi, Pescara, and Rimini), and one Italian port (i.e., Ancona).

In the short term, behavioural changes produced modal shifts and variations in the total amount of passengers. These changes may be useful for understanding the potential impact of mid- and long-term structural shifts. For example, the majority (around 73%) of respondents used public transport when travelling in the pre-COVID-19 period, whereas the corresponding share of respondents in the COVID-19 period was 46%, and approximately 56% in the post-COVID-19 future. The main reason for not travelling by public transport was related to personal or sanitary reasons in the COVID-19 period. During the pandemic, around 39% of travellers preferred their own vehicle to public transport for moving to/from the airports and ports because of safety and sanitary reasons.

A decrease in the volume of passenger traffic will likely act as a buffer for the higher external costs related to a shift in passengers from public transport to private vehicles going to/from the airports of Rimini, Dubrovnik, and Pula, and the port of Dubrovnik. Attention should be paid to potential changes in the habits and trends of travellers to modulate the offer of the modes of transport to/from the airport/port infrastructures, particularly for the locations (e.g., Pula, Bari, Brindisi, and Pescara) served by airports and ports with volumes of passenger traffic higher than or comparable to the pre-COVID-19 levels.

With the pandemic, health was the main reason behind daily choices for up to 49% of the respondents. Moreover, a similar share of travellers considered health before choosing transportation during and after the pandemic. Travellers' concerns for health could be the key for boosting the transition to smart and sustainable modes of transport.

The findings of the present research may be useful to policy makers, stakeholders, and marketing analysts for tailoring transport modes to the behaviours and needs of travellers departing from small- to medium-sized airports and ports. This survey method can be easily replicated for investigating changes in consumer habits deriving from a future crisis with a psychological burden on the population and demand-side disruption, such as the situation represented by the COVID-19 pandemic. Future research dealing with surface access to airport and port infrastructures should consider airport and port authority surveys for identifying new challenges deriving from (i) abrupt changes in passenger volumes and behaviour, (ii) cooperation with the authorities and stakeholders involved in passenger transport, and (iii) emerging trends and future policies about sustainable and smart mobility.

**Author Contributions:** Conceptualization, E.M., G.P. and S.V.; methodology, E.M.; formal analysis, E.M., S.V. and U.R.; investigation, E.M., G.P. and S.V.; resources, G.P.; data curation, S.V.; writing—original draft preparation, E.M.; writing—review and editing, E.M., G.P., U.R., F.C. and A.G.; visualization, E.M.; project administration, G.P., F.C. and A.G.; funding acquisition, F.C. All authors have read and agreed to the published version of the manuscript.

**Funding:** This research was funded by INTERREG, grant number 10044741, for the Green and Intermodal Solutions for Adriatic Airports and Ports (ADRIGREEN) project under the Interreg V-A Italy Croatia.

**Institutional Review Board Statement:** Not applicable.

**Informed Consent Statement:** Not applicable.

**Data Availability Statement:** Datasets can be made available upon requests and on a case-by-case basis provided the anonymity of the data.

**Acknowledgments:** The results and opinions presented in this paper are those of the authors. The authors are grateful to the airports' and ports' team members who helped in organizing the survey at the terminals. Enrico dedicates his work to his beloved Olga, for her loving kindness.

**Conflicts of Interest:** The authors declare no conflict of interest.

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
