# Peer review of "New Habits of Travellers Deriving from COVID-19 Pandemic: A Survey in Ports and Airports of the Adriatic Region"

_sustainability, doi:10.3390/su14148290_

Round 1

Reviewer 1 Report

Excellent contribution. Well done!

Please provide one style trough the paper 24.5 and 30.3 % VERSUS 12 %.

Learning more about the need of further research would improve the publication. I am also wondering if any recommendations can be formulated for upcoming crisis.

Author Response

To Reviewer 1,

please, see the attached file. 

Reviewer 2 Report

Introduction

the introduction and the methodology of the study need to be improved

The manuscript would benefit from English language editing. There are also some typographical errors.

The authors should highlight the main contribution of the work to the literature.

The objective still not clear, please try to explain more the objective and to include the motivations behind this work

Authors need to improve the introduction and including more studies/articles in the litterature review

By the end of the introduction, the authors should describe the paper structure.

Data and calculations

Could you please explain more the survey process, sample definition
I thinks that the reader can be confused between flyers or is it an online survey ? or the flyer is only to invite people to answer the questionnaire

I would like to have a justification for choosing this specific area and adding a map to better identify the study area would be an asset

table 1: it would be better to calcuate the decrease rate

line 135: what about the number of flights is it the same ?

line 153: Explain more you eliminated 32 answers, which criteria did you considered

Conclusion

Results are poorly discussed and associated with some implications and recommendations for using such a method. I feel that this part needs further discussion and should be rewritten.

Author Response

To Reviewer 2, 

please, see the attached file.

Reviewer 3 Report

This paper gives some analysis about the influence of COVID-19 in the traveler habits. Some comments are listed as follows:

1. Obviously, COVID-19 would affect the choice of traveler habits. To avoid infection, most people would use their own vehicles. So, please give more descriptions about the motivation and necessity of this research.

2. What is the significance of this research? Please also make this part clear.

Author Response

To Reviewer 3, 

please, see the attached file. 

Round 2

Reviewer 2 Report

Dear authors, 

The paper has improved substantially. Congratulation to all authors. I hope you can improve the layout of all inserted figures and tables before the publication 

Best regards, 

Author Response

To Reviewer 2, 

We thank the referee for the comments and suggestions. Please, find attached an improved version of the layout of the manuscript. 
